# Anti-Bacterial and Anti-Biofilm Activities of Essential Oil from *Citrus reticulata* Blanco cv. Tankan Peel Against *Listeria monocytogenes*

**DOI:** 10.3390/foods13233841

**Published:** 2024-11-28

**Authors:** Jinming Peng, Guangwei Chen, Shaoxin Guo, Ziyuan Lin, Yue Zeng, Jie Ren, Qin Wang, Wenhua Yang, Yongqian Liang, Jun Li

**Affiliations:** 1Guangdong Key Laboratory of Science and Technology of Lingnan Specialty Food, Guangzhou 510225, China; pengjmiyz@163.com (J.P.); wangqin@zhku.edu.cn (G.C.); why10000why@126.com (W.Y.); 2Key Laboratory of Green Processing and Intelligent Manufacturing of Lingnan Specialty Food, Ministry of Agriculture, Zhongkai University of Agriculture and Engineering, Guangzhou 510225, China; 3Nuspower Greatsun (Guangdong) Biotechnology Co., Ltd., Guangzhou 510900, China; renjiescut2001@163.com; 4School of Pharmacy, Guangdong Pharmaceutical university, Guangzhou 510006, China

**Keywords:** *Listeria monocytogenes*, essential oils, cell membrane, biofilm, extracellular polymeric substances, transcriptome analysis

## Abstract

In recent years, plant essential oils have been confirmed as natural inhibitors of foodborne pathogens. *Citrus reticulata* Blanco cv. Tankan peel essential oil (CPEO) showed anti-*Listeria monocytogenes* (LM) activities, and this study investigated the associated mechanisms by using high-resolution electron microscope, fluorescence spectrometer, flow cytometer, potentiometer, and transcriptome sequencing. The results showed that CPEO restrained LM growth at a minimum inhibitory concentration of 2% (*v/v*). The anti-LM abilities of CPEO were achieved by disrupting the permeability of the cell wall, damaging the permeability, fluidity, and integrity of the cell membrane, disturbing the membrane hydrophobic core, and destroying the membrane protein conformation. Moreover, CPEO could significantly inhibit the LM aggregation from forming biofilm by reducing the extracellular polymeric substances’ (protein, polysaccharide, and eDNA) production and bacterial surface charge numbers. The RNA sequencing data indicated that LM genes involved in cell wall and membrane biosynthesis, DNA replication and repair, quorum sensing and two-component systems were expressed differently after CPEO treatment. These results suggested that CPEO could be used as a novel anti-LM agent and green preservative in the food sector. Further studies are needed to verify the anti-LM activities of CPEO in real food.

## 1. Introduction

*Listeria monocytogenes* (LM), one of the four major foodborne pathogens, is widely found in diversified contaminated foods, such as seafood, meat, raw milk, fruits, and vegetables [1]. LM can form biofilms to enhance its viability in adverse environments, for example, low temperature, nutrient deficiency, and pH change, etc. [2]. Ingesting LM-laden food can lead to listeriosis, a severe foodborne illness with a high fatality rate (around 30%), particularly affecting susceptible populations, including the elderly, pregnant women, and those with underlying health problems [3]. Hence, developing anti-LM agents is of great significance for ensuring food safety and public health. Because the utilization of synthetic anti-bacterial factors easily causes bacterial resistance and environmental pollution [4], natural anti-LM substances, including polyphenols, essential oils, and peptides, have recently garnered significant attention due to their multifaceted advantages [5,6,7].

Citrus is one of the main processed fruits in the world, producing massive amounts of waste, including peels, seeds, and membrane residue every year [8]. Citrus peel waste comprises nearly 50–55% of the wet weight in whole fruit [9]. Currently, essential oils from citrus fruit peel have attracted extensive attention because of their high yields, strong aromas, and especially broad-spectrum anti-bacterial properties [10,11]. Citrus essential oils are widely used as a natural additive in food processing because they have been identified as safe by the FDA [12]. *Citrus reticulata* Blanco cv. Tankan is a high-quality citrus variety that has been cultivated in China for 1300 years [13]. Like other citruses, *Citrus reticulata* Blanco cv. Tankan fruit peel as the major by-product is often discarded or used as animal feed, fertilizer, and fuel [14]. Previous research has discovered that *Citrus reticulata* Blanco cv. Tankan peel essential oil (CPEO) could strongly inhibit foodborne pathogens *Staphylococcus aureus* and *Escherichia coli*, with a minimum inhibitory concentration (MIC) value of 2.5 μg/mL [13]. However, to our knowledge, the anti-bacterial and anti-biofilm activities and mechanisms of CPEO on LM have not yet been studied.

Therefore, the anti-bacterial activities of CPEO against LM were evaluated by examining its influences on the cell morphology, membrane properties, and membrane protein conformation. Additionally, the impacts of CPEO treatment on LM biofilm was investigated by measuring the biofilm biomass, bacterial aggregation ability, extracellular polymeric substance (EPS) production levels, and bacterial surface charge. Finally, transcriptome analysis was adopted to elucidate the anti-LM mechanism of CPEO at the genetic level.

## 2. Materials and Methods

### 2.1. Materials

*Citrus reticulata* Blanco cv. Tankan peel was obtained from Guangdong Plum Industry Food Ltd. (Raoping, China) in January 2023. Brain Heart Infusion (BHI) broth was obtained from Huankai Biology (Zhaoqing, China). The analytical reagents were from Sinopharm Chemical Reagent (Shanghai, China). 1-Palmitoyl−2-oleoyl-sn-glycero−3-phosphatidylglycerol (POPG), N−(7-nitrobenz−2-oxa−1,3-diazol−4-yl)−1,2-dihexadecanoyl-sn-glycero−3-phosphoethanolamine triethylammonium salt (NBD-PE), 1−[4−(trimethylammonio)phenyl]−6-phenyl−1,3,5-hexatriene (TMA-DPH), and 1,6-diphenyl−1,3,5-hexatriene (DPH) were from Sigma-Aldrich (St Louis, MO, USA).

### 2.2. CPEO Extraction

Fresh *Citrus reticulata* Blanco cv. Tankan peel (0.5 kg) was sectioned and subjected to steam distillation in purified water (2 L) for 3 h; the essential oils was then harvested and dehydrated using anhydrous sodium sulfate. CPEO was kept at 4 °C in a dark place.

### 2.3. CPEO Composition Analysis

GC-MS analysis of CPEO was conducted by using an Agilent 7890A GC system equipped with a flame ionization detector as mentioned previously [15]. The separation was carried out using an RTX−5MS capillary column (5% phenylmethyl-polysiloxane, with the specification of 30 m × 0.25 mm × 0.25 µm). The oven was held at 80 °C for 3 min after injection, then ramped to 260 °C at 3 °C/min, and held at 260 °C for 15 min. The split ratio was 1:10, and the electron ionization voltage was 70 eV. Volatile components in the CPEO were identified using the National Institute of Standards and Technology’s (NIST) mass spectra library and document data.

### 2.4. Minimum Inhibitory Concentration (MIC) and Time Kill Analysis

*Listeria monocytogenes* (ATCC19115) was cultured in BHI media with shaking for 48 h at 37 °C. The CPEO was diluted into a BHI medium at a serial two-fold (0.5–16%, *v/v*) containing 1 × 10^6^ CFU/mL LM. The MIC was the minimum concentration of CPEO in which the medium kept clear. For the time kill experiments, LM (1 × 10^6^ CFU/mL) was intervened with CPEO at the concentrations of 1%, 2%, and 4% (*v/v*). The suspension without CPEO served as a control group (CON). During the 24 h incubation period, the sample was then incubated at 37 °C with shaking. The sample absorbance at OD_600 nm_ was read at various time points (0, 1, 2, 4, 8, 16, and 24 h).

### 2.5. Extracellular Alkaline Phosphatase (AKP) Activity Determination

LM (1 × 10^8^ CFU/mL) was cultured in BHI media containing CPEO at concentrations of 1%, 2%, and 4% (*v/v*) with shaking for 3 h at 37 °C. The culture supernatants were harvested by centrifugation (5000 rpm, 10 min), and its extracellular AKPase was measured by using the alkaline phosphatase activity assay kit (Solarbio, Beijing, China).

### 2.6. Cell Membrane Properties Analysis in LM

#### 2.6.1. Electron Microscopy Observation

LM shape and structure were analyzed via scanning electron microscope (SEM) and a transmission electron microscope (TEM). LM (1 × 10^7^ CFU/mL) was treated with CPEO at the concentration of 2% (*v*/*v*) for 3 h. The culture broth without CPEO served as a control (CON). The collected bacteria were fixed overnight in 2.5% (*w*/*v*) glutaraldehyde at 4 °C, then washed three times with PBS. The cell pellet was dehydrated with gradient ethanol (30–100%) for 20 min of treatment per concentration. The sample was lyophilized and coated with gold, and then visualized by using SU−8010 SEM (HITACHI, Tokyo, Japan). For TEM, the bacteria cells were negatively stained with 1% phosphotungstic acid for 5 min, and then observed by using HT−7700 TEM (HITACHI, Tokyo, Japan).

#### 2.6.2. Fluorescence Polarization Measurement

The polarity-reactive fluorescent probe DPH was employed to measure the fluctuations in membrane fluidity; the larger the *P* value, the smaller the membrane fluidity. LM suspensions (1 × 10^8^ CFU/mL) were incubated with CPEO at the concentrations of 1%, 2%, and 4% (*v/v*) for 3 h at 37 °C. The culture broth without CPEO served as the control (CON). After that, the bacteria were labeled by the fluorescent probe DPH (1 μM), and then resuspended in PBS. The fluorescence strength (Ex = 370 nm, Em = 426 nm) was detected by using a F−4600 fluorescence spectrophotometer equipped with a polarizer. The polarization (*P*) value was computed with the subsequent equation
$$P=\frac{I_{0,0}-GI_{0,90}}{I_{0,0}+GI_{0,90}};G=\frac{I_{90,0}}{I_{90,90}};$$
where *I* represents the fluorescent intensity of the emitted beam in various directions. 0 and 90 indicate the vertical and horizontal polarizer orientations, respectively. *G* means the grating correction factor.

#### 2.6.3. Flow Cytometry Analysis

The cell membrane integrity of LM was determined by using flow cytometry, as mentioned previously [16]. The bacterial cells (1 × 10^8^ CFU/mL) were treated with or without the CPEO at the concentrations of 1%, 2%, and 4% (*v/v*). After 24 h incubation, the cells from each intervention group were harvested with PBS (0.1 M), and further dyed with propidium iodide (5 mM) for 15 min. The fluorescent strength was detected by the AccuriTM C6 Plus flow cytometer (BD Biosciences, Franklin Lakes, NJ, USA).

#### 2.6.4. Fluorescence Spectrum Analysis

As previously described [17], the effects of CPEO on the LM membrane protein were assessed by analyzing the changes in the fluorescence spectra of certain amino acids. Briefly, PBS-suspended LMs (1 × 10^8^ CFU/mL) were incubated with CPEO at the concentrations of 1%, 2%, and 4% (*v/v*) for 2 h at 37 °C. The CPEO-free group was defined as the control (CON). The fluorescent tyrosine (Ex = 296 nm, Em = 280–450 nm), tryptophan (Ex = 280 nm, Em = 280–400 nm), and phenylalanine (Ex = 258 nm, Em = 280–400 nm) in the membrane protein were determined by using a plate reader.

### 2.7. CPEO–Membrane Interactions in POPG Liposome

#### 2.7.1. Liposome Preparation

The complexity of the LM cell membrane’s structure made it difficult to explore the mechanisms of the membrane–compound interactions. Currently, liposomes are commonly used as a good model for studying how substances affect the cell membrane. Previous studies reported that the bacterial cell membrane contains a substantial number of anionic phospholipids, especially phosphatidylglycerol (PG) [18]; POPG liposome was therefore selected to model the LM cell membrane in this research. The interactions between CPEO and the liposomal membrane were assessed through fluorescence spectrometry utilizing various fluorescent probes (NBD-PE, TMA-DPH, and DPH). NBD-PE, a surface-sensitive probe, can monitor changes on the bilayer lipid–water interface. TMA-DPH is an amphiphilic probe that is located on the lipid headgroup region. DPH, a lipophilic fluorophore, can detect variations near the bilayer hydrophobic core [19]. The liposome was produced via thin-film hydration sonication as described previously [20]. Briefly, POPG was solubilized as a methanol–chloroform (3:7, *v/v*) solvent system-containing a fluorescent probe (1 μM NBD-PE/TMA-DPH/DPH). The mixture was then evaporated until the thin film appeared. Subsequently, the film was suspended with PBS via ultrasound for 15 min. Finally, the mixed solution was extruded 20 times using an extruder with a 100 nm polycarbonate membrane to acquire the fluorescent liposome.

#### 2.7.2. Fluorescent Strength Determination

The different fluorescence liposome was separately mixed with CPEO at 37 °C for 30 min. The untreated group served as a control (CON). The fluorescent spectra (Ex = 468 nm, Em = 500–600 nm) of the NBD-PE-tagged liposome was scanned, and the spectrum (Ex = 360 nm, Em = 380–500 nm) of the TMA-DPH/DPH-labeled liposome was recorded by using a plate reader.

### 2.8. Biofilm Formation Analysis

#### 2.8.1. Biofilm Biomass Determination

LM (1 × 10^7^ CFU/mL) was cultured in 96-well plates with or without CPEO for 96 h. At the corresponding time, the biofilms were colored with 0.1% (*v/v*) crystal violet for 15 min, and then washed with PBS. Thereafter, the attached dye was dissolved with 95% ethanol, and then recorded at OD_570 nm_.

#### 2.8.2. Bacterial Aggregation Ability

LM (1 × 10^7^ CFU/mL) was incubated with CPEO at 37 °C for 24 h. The bacteria suspension without CPEO served as the control group (CON). After incubation, the absorbance (A_1_) of the static bacterial solution was measured at OD_600 nm_. Each group was then suspended, with the absorbance (A_2_) of the suspended bacteria further detected at OD_600 nm_. The aggregation rate was computed as follows: % aggregation rate = [(A_2_ − A_1_)/A_1_] × 100%.

#### 2.8.3. EPS Determination

LM (1 × 10^7^ CFU/mL) was cultured with or without CPEO at 37 °C for 24 h. Biofilm samples were harvested with ultrapure water at 4 °C. The mixture was disrupted via ultrasound for 1 min, followed by centrifugation (12,000 rpm, 10 min). The supernatant was passed through a 0.45 μm filter to collect the EPS. The protein level in the EPS was detected by a BCA-protein quantification kit (Servicebio, Wuhan, China). The polysaccharide content in the EPS was quantified using the phenol–sulfuric acid method [21]. The eDNA was enriched from the EPS by the DNA extraction kit (Solarbio, Beijing, China), and then quantified at OD_260 nm_.

#### 2.8.4. Bacterial Zeta Potential Measurement

LM (1 × 10^7^ CFU/mL) was cultured with or without CPEO at 37 °C for 24 h. For the assay, the bacteria cells were mixed in PBS (10 mM, pH = 7.0) to achieve an optical density of 0.8 at 600 nm. The zeta potential of the free-floating bacteria was measured using a Zetasizer Nano ZSP (Malvern Instruments, Malvern, UK), and each sample was tested three times.

### 2.9. Transcriptome Analysis

A transcriptome analysis was performed as previously mentioned [22]. The LMd (1 × 10^10^ CFU/mL) were treated with or without the CPEO at the concentration of 1% (*v*/*v*) for 48 h at 37 °C. The total RNA was collected via a TRIzol-based procedure (Life Technologies, Carlsbad, CA, USA), and then purified via agarose gel electrophoresis. The purity and yield of the RNA were evaluated via a NanoDrop™ One spectrophotometer (Thermo Fisher Scientific, Hillsboro, OR, USA) and Bioanalyzer 2100 (Agilent Technologies, USA). Each sample library was sequenced on the Illumina platform (Genedenovo, China). Gene ontology (GO) enrichment enriched the GO terms in the DEGs and filtered the DEGs corresponding to three ontologies: molecular function, cellular component, and biological process. A DEGs-enriched signal transduction pathway was identified through the Kyoto Encyclopedia of Genes and Genomes’ (KEGG) enrichment.

### 2.10. Statistical Analysis

The data were analyzed, employing GraphPad Prism 10 and SPSS Statistics 28. The results were expressed as means ± SD from three independent experiments. The statistical significance (*p* < 0.05) was determined using a one-way analysis of variance (ANOVA), followed by Tukey’s HSD test.

## 3. Results and Discussion

### 3.1. Chemical Compositions of CPEO

The CPEO yield obtained from hydrodistillation was 0.332% (*w*/*w*). The GC-MS profiling of the CPEO compositions is listed in Table 1. Six compounds were identified from CPEO. The main component in CPEO was limonene (95.993%). The other compounds were identified as β-myrcene (2.689%), α-pinene (0.953%), α-sabinene (0.158%), α-fenchene (0.151%), and α-phellandrene (0.056%). Yang et al. [23] found that limonene (50.88–97.19%) was an abundant component in all the essential oils from 21 citrus cultivars, followed by γ-terpinene, β-myrcene, α-terpineol, and β-pinene. In comparison, the limonene level of CPEO was higher than that of the *Citrus limon* (L.) Osbeck ‘Lisbon’ essential oils (69.01%) and the *Citrus maxima* (Burm.) Merr. essential oils (76.33%) [23].

### 3.2. Anti-Bacterial Activity of CPEO

As seen in Figure 1A, no prominent (*p* > 0.05) growth was observed at concentrations above 1% (*v*/*v*); thus, the MIC value of CPEO against LM was 2% (*v/v*). Previous studies [24,25] found that the MIC value of the essential oils obtained from Citrus Changshan-huyou Y.B. Chang or Fingered Citron against LM was 4% (*v/v*), which confirmed the strong inhibitory ability of CPEO against LM. The growth curves of LM were further compared before and after CPEO intervention. Figure 1B shows the LM growth was concentration-dependently suppressed by CPEO during a 24 h incubation period.

### 3.3. SEM and TEM Analysis of LM Treated with CPEO

The morphology and structure of the LM treated with or without CPEO at 2% (*v*/*v*) were investigated in this study. SEM analysis showed that the untreated LM appeared to have normal membrane surfaces, with a complete and smooth clavate boundary (Figure 2A). After treatment with CPEO, the surfaces of some of the bacteria became rough, collapsed, and shrunken (Figure 2B), indicating that CPEO possessed a serious detrimental ability on the morphology of the cell membrane. Additionally, the intracellular ultra-structure of LM was further visualized by TEM. As seen in Figure 2C,D, cell lysis, membrane rupture, and intracellular substances leakage were also observed in the treatment group, while the bacteria remained intact in the non-treated group, suggesting that CPEO had great destructive effects on the cell walls. This could be because liposoluble CPEO strongly interacted with highly hydrophobic cell walls to destroy its original structure [26]. These findings were in accordance with earlier reports, which indicated that the essential oils derived from *Citrus limon* var *pompia* and *Melissa officinalis* could disrupt the cell wall and cell membrane of LM [27,28].

### 3.4. Influences of CPEO Treatment on the Cell Wall and Cell Membrane of LM

The cell wall and cell membrane are the indispensable barriers for bacterial cells to resist the harsh environment, which play the role of material exchange, signal transduction, and energy transmission [29]. This hinted that the cell barrier is regarded as the potential target of essential oils. Firstly, the influences of CPEO on the cell wall permeability of LM were evaluated by measuring the extracellular alkaline phosphatase (AKP) levels in this study. AKP was released into the extracellular environment when the bacterial cell wall was damaged [30]. Figure 3A presents that the AKPase levels in LM that intervened with CPEO at concentrations of 1%, 2%, and 4% (*v/v*) were 1.13 U/L, 1.20 U/L, 1.46 U/L, respectively, which manifested a substantial rise in comparison to the control group (0.73 U/L). The above results suggested that CPEO could concentration-dependently destroy LM cell wall permeability.

Next, the effects of CPEO on the permeability, fluidity, and integrity of the LM cell membrane were further investigated in this study. To explore whether CPEO disrupts the cytomembrane permeability, the leakage of the intracellular proteins from LM was measured. Figure 3B shows that the intracellular protein level remarkably (*p* < 0.05) increased as the concentration of CPEO grew from 1% to 4% (*v/v*). Consistent with this study, Li et al. also observed a substantial increase in LM cell membrane permeability, accompanied by elevated protein and DNA leakage after basil essential oil treatment [31]. The influences of CPEO on the membrane fluidity of the bacteria cells was examined through DPH fluorescent polarimetry. Figure 3C presents that the *P* values of the DPH-tagged bacteria were distinctly (*p* < 0.05) increased with the concentrations of CPEO increasing, suggesting that CPEO could evidently decrease the LM membrane fluidity. Previous studies confirmed that maintaining cytoplasmic membrane fluidity is essential for LM growth [32]. Thus, it was plausible that the anti-LM activities of CPEO were achieved by decreasing the membrane fluidity. Propidium iodide (PI), a fluorescence probe, can specifically cross the impaired cells and combine with intracellular DNA, leading to increased fluorescence strength [33]. Compared with the control (5.9%), the percentages of PI-stained cells (V1-R) were 59.8%, 97.9%, and 98.8% after exposure to CPEO at 1%, 2%, and 4% (*v/v*), respectively (Figure 3D). Similar experimental results were also observed in the cinnamon essential oils [34]. This suggests that CPEO could intensely damage the integrity of the LM cell membrane.

### 3.5. Membrane-Disturbing Effects of CPEO in a POPG Model

Figure 4A–C shows that CPEO could dose-dependently decrease the fluorescent strengths of NBD-PE, TMA-DPH, and DPH. This indicated that CPEO could disrupt the cell membrane at various sites including the lipid–water interface, the lipid polar headgroup region, and the hydrophobic core. CPEO’s abilities to quench various fluorescent probes were observed to follow the order of DPH > TMA-DPH > NBD-PE (Figure 4D). The quenching rate of CPEO at 4% (*v/v*) for NBD-PE, TMA-DPH, and DPH fluorescence probes was 13.43%, 18.44%, and 26.76%, respectively. This demonstrated that CPEO was located mainly on the hydrophobic core of the bilayer, followed by the polar headgroup and the water–lipid interface of the bilayer. This might have been due to the strong lipid-soluble substance limonene (the main component in CPEO) interacting with the hydrophobic tails of the phospholipids to destroy the membrane structure [35]. Therefore, CPEO might have exhibited strong anti-LM activity by disrupting the intra-membranous hydrophobic core.

### 3.6. Membrane Protein Conformation of LM Treated with CPEO

Amino acid residues, tyrosine (Tyr), tryptophan (Trp), and phenylalanine (Phe) serve as key fluorophore in membrane proteins [36]. As seen in Figure 5A, the Tyr residues reached a maximal fluorescent peak at 294 nm; a decrease in fluorescence signal in the CPEO-treated group was observed in a dose-dependent manner with the concentrations increasing from 1% to 4% (*v/v*), compared to the untreated group (CON). The Tyr fluorescent quenching rate of CPEO at 4% (*v/v*) was 20.30%. Analogously, the characteristic spectrum of Trp and Phe also manifested that CPEO had strong fluorescent quenching abilities. Figure 5B,C show how CPEO dose-dependently quenched the characteristic spectrum of Trp and Phe. After treatment with CPEO at 4% (*v/v*), the fluorescence quenching rate of Trp and Phe was 67.37% and 67.58%, respectively. By contrast, CPEO possessed stronger quenching abilities on Trp and Phe than Tyr. These results suggested that CPEO could change LM membrane protein conformation, resulting in cell membrane impairment.

### 3.7. Inhibition of LM Biofilm Formation by CPEO

The LM’s persistence in the harsh environment is attributed to its ability to form biofilms [37]. This study further evaluated the anti-biofilm activities of CPEO against LM via a crystal violet staining assay. The higher the OD value was, the more biofilm mass there was. Figure 6A shows that the OD values rapidly increased with time, and became stable after 72 h in the control group. However, the OD values for the CPEO treatment groups at 2% and 4% (*v/v*) were significantly lower compared to the control group (CON), suggesting that CPEO could significantly restrain LM biofilm formation. Aggregation has been established as a critical mechanism for bacterial adhesion and biofilm formation [38]. Figure 6B shows that CPEO dose-dependently decreased the aggregation rate of planktonic LM, which was in agreement with Zhang et al., who evaluated the anti-biofilm efficacy of clove essential oils against LM [39]. Extracellular polymeric substances (EPS), predominantly make up of proteins, polysaccharides, lipids, and nucleic acids, are crucially important for the adhesion and biofilm formation of LM [40]. Figure 6C shows that the levels of protein, polysaccharide, and eDNA in the CPEO-treated groups were markedly decreased (*p* < 0.05) compared to the untreated group (CON). The protein, polysaccharide, and eDNA content in the EPS was reduced by 47.77%, 54.74%, and 63.74%, respectively, in the 2% (*v*/*v*) CPEO group. Furthermore, it has been confirmed that the LM surface contained abundant negatively charged groups (carboxyl and phosphoric acid groups), and the large number of negative charges helped the planktonic bacteria aggregate and adhere to the biotic or abiotic interface through electrostatic interaction, thus promoting the formation of biofilm [41]. Figure 6D shows that the zeta potential of the bacterial suspension maintained a stable value (approximately −20.23 mV) in the control group (CON). However, CPEO reduced the negative charge of LM in a dose− and time-dependent manner. During incubation from 15 to 90 min, the negative charge decreased from −18.64 to −14.14 mV in the 2% (*v*/*v*) CPEO group and from −18.54 to −10.26 mV in the 4% (*v*/*v*) CPEO group. Recently, protocatechuic acid and vanillic acid were proven to antagonize the adhesion of *Escherichia coli* to form biofilms by reducing the surface charge number [42]. Therefore, the anti-biofilm activity of CPEO might have been achieved by inhibiting the bacterial aggregation; this inhibition was likely mediated by reducing the biofilm EPS production and bacterial surface charge number.

### 3.8. Transcriptome Analysis of LM Treated with CPEO

Differentially expressed genes (DEGs) were identified in the CPEO-treated group versus the untreated group (CON). Figure 7A,B presents that 1207 DEGs were found in LM, with 430 up-regulations and 777 down-regulations (|log_2_ (fold change)| > 1, false discovery rate < 0.05). The large number of DEGs in LM indicated a strong cellular sensitivity to the CPEO. Hence, the subsequent analysis primarily concentrated on LM’s DEGs to elucidate the anti-bacterial mechanism at the genetic level.

Next, GO and KEGG enrichment analyses were executed to uncover the functions, metabolic pathways, and interactions of the DEGs associated with CPEO’s anti-LM activities. The 239 GO terms encompassed the 1207 DEGs found in LM, comprising 110 terms for the biological process (BP), 110 terms for the molecular function (MF), and 19 terms for the cellular component (CC). Among them, the cellular process (GO:0009987), the metabolic process (GO:0008152), the single-organism process (GO:0044699), localization (GO:0051179), biological regulation (GO:0065007), the response to stimulus (GO:0050896), and cellular component organization or biogenesis (GO:0071840) were the predominant terms in BP ontology. For MF ontology, the primary terms were concentrated at the catalytic activity (GO:0003824), binding (GO:0005488), and transporter activity (GO:0005215). For CC ontology, the leading terms were gathered at the cell (GO:0005623), cell part (GO:0044464), membrane (GO:0016020), membrane part (GO:0044425), and macromolecular complex (GO:0032991). Furthermore, KEGG analysis was applied to the group DEGs into specific pathways, shedding light on their functions and interactions. Figure 7D shows that the nucleotide excision repair pathway was the top enriched pathway among the DEGs, followed by phenylalanine metabolism, aminoacyl-tRNA biosynthesis, ABC transporters pathways, etc. This finding partly demonstrated that the anti-LM capacities of CPEO were mediated by the aforementioned pathways.

#### 3.8.1. DEGs Associated with Cell Structures

The cell wall and membrane are essential for maintaining the fine structure of bacteria, due to their components, including peptidoglycan, lipopolysaccharides, and lipid bilayer [43]. Consequently, it is essential to screen the DEGs associated with cell wall synthesis and cell membrane integrity. Figure 8 shows that both the cell wall and cell membrane of LM were significantly impacted by CPEO. In comparison with the control group (CON), eight genes related to peptidoglycan biosynthesis (ko00550) were significantly lowered in the CPEO-intervened group compared to the CON. Multiple genes associated with the cell wall metabolic pathways were prominently down-regulated in response to CPEO, including *pgmB* and *celB* for starch and sucrose metabolism (ko00500), respectively, and *manX* and *manA* for amino sugar and nucleotide sugar metabolism (ko00520), respectively. Moreover, CPEO decreased the expressions of the cell membrane-related DEGs including *dhaK*, *dhaL*, *plsY*, *dhaM*, *glpK*, *dhaL*, and *plsX* for glycerolipid metabolism (ko00561), *eutA*, *cls*, and *plsY* for glycerophospholipid metabolism (ko00564). Therefore, the anti-LM mechanism of CPEO was achieved by inhibiting the synthesis of the cell wall and the cell membrane components, and destroying the integrity of the cell wall and cell membrane.

#### 3.8.2. DEGs Associated with DNA Replication and Repair

The processes of DNA replication and repair are essential for the survival of microorganisms [44]. The majority of bacterial species necessitate the biosynthesis and/or uptake of purines and pyrimidines, which are the fundamental building blocks of nucleotides, for their survival [45]. The DEGs linked to DNA replication and the repair systems (homologous recombination, nucleotide excision repair, DNA replication, pyrimidine metabolism, and purine metabolism) were obtained after comparing the treatment and control samples (Figure 9). Among them, 9 of the 10 significantly down-regulated genes (*recR*, *holB*, *recA*, *dnaE*, *recF*, *holA*, *ruvA*, A4W89_RS02745, *priA*, and *recO*) involved in homologous recombination (ko03440) were observed in the CPEO-treated group. The DEGs (*pcrA*, *mfd*, *ligA*, *uvrA*, *uvrC*, and *uvrB*) associated with nucleotide excision repair (ko03420) were down-regulated upon exposure to CPEO. The DEGs (*dnaE*, *ligA*, *holB*, *holA*, and *dnaB*) associated with DNA replication (ko03030) were down-regulated after CPEO intervention. Furthermore, it was found that CPEO down-regulated DNA synthesis was related to the DEGs, including *nrdD*, *carB*, and *carA* for pyrimidine metabolism (ko00240), and *purD*, *purN*, *nrdD*, and *purH* for purine metabolism (ko00230). The above results suggested that the DNA replication and repair systems were destroyed after CPEO intervention, which inhibited the protein biosynthesis and metabolism, therefore suppressing LM growth.

#### 3.8.3. DEGs Linked to Biofilm Formation

Quorum sensing (QS), a ubiquitous bacterial cell–cell communication system, generates autoinducers which promote planktonic LM aggregation and biofilm formation [46]. The two-component system (TCS) manages the main constituents of biofilm formation in reaction to environmental stimulation, causing LM to change from a free-floating to biofilm state [47]. Upon exposure to CPEO, the transcriptional profiles of genes implicated in the QS system and the TCS were significantly modulated in the planktonic LM cells (Figure 10). In contrast to the control group (CON), numerous genes (*ftsY*, *secA*, *hly*, and *trpE*) associated with the QS system were down-regulated upon exposure to the CPEO. Robert et al. uncovered that *hly* inactivation significantly reduced LM aggregation and biofilm formation [48]. Moreover, it was found that the TCS related to the DEGs (*dhaA*, *dltA*, *glnA*, and *dltD*) were down-regulated after CPEO treatment. Alonso found that the deletion of *dltA* and *dltD* could decrease the extracellular amino acid level and disturb the surface charge and lipoteichoic acid thickness of LM, thereby inhibiting the bacteria adhesion to form biofilm [49]. Therefore, this study revealed that CPEO could inhibit LM biofilm via the down-regulating genes associated with aggregation ability and surface property.

## 4. Conclusions

In the current study, CPEO was demonstrated to concentration-dependently suppress LM with a minimum inhibitory concentration of 2% (*v/v*). The anti-LM activity of CPEO was achieved by disrupting the cell wall permeability, damaging the permeability, fluidity, and integrity of the cell membrane, and destroying the membrane protein conformation. The liposome model analysis demonstrated that CPEO could strongly disturb the hydrophobic core of the cell membrane interior. Furthermore, the CPEO was proved to inhibit the LM aggregation to form biofilm by reducing EPS (protein, polysaccharide, and eDNA) production and bacterial surface charge number. A transcriptome analysis further confirmed the aforementioned conclusion at the genetic level. The DEGs associated with the cell wall and membrane biosynthesis, DNA replication and repair, and quorum sensing and two-component systems were remarkably impacted in the planktonic LM after CPEO treatment. Therefore, CPEO has the potential to be used as a natural antimicrobial agent to control LM in the food industry. Further research is required to investigate the anti-LM effects of CPEO in actual food systems.

## Figures and Tables

**Figure 1 foods-13-03841-f001:**
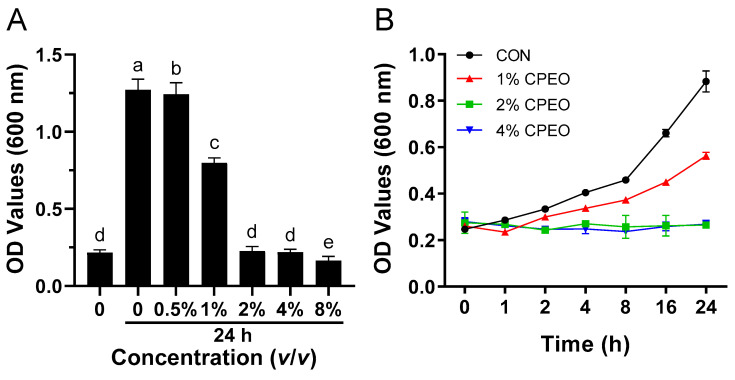
Minimum inhibitory concentration of *Citrus reticulata* Blanco cv. Tankan peel essential oil (CPEO) on LM (**A**). Growth curvatures of LM with and without exposure to CPEO (**B**). Columns marked with different letters imply that the differences are statistically significant (*p* < 0.05).

**Figure 2 foods-13-03841-f002:**
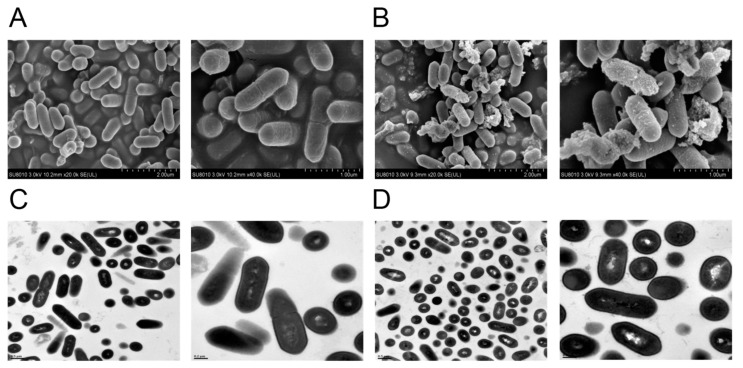
SEM micrographs of untreated LM (**A**) and *Citrus reticulata* Blanco cv. Tankan peel essential oil (CPEO)-treated LM (**B**). TEM images of untreated LM (**C**) and CPEO-treated LM (**D**).

**Figure 3 foods-13-03841-f003:**
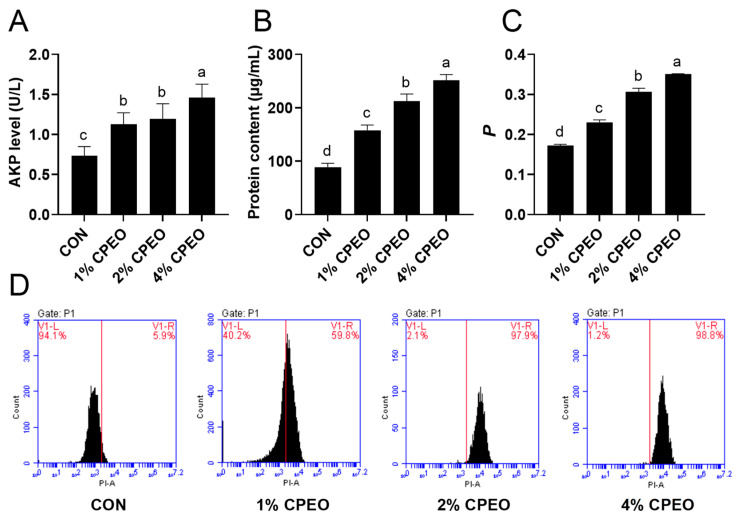
Alkaline phosphatase (AKP) (**A**) and protein (**B**) leakage from LM treated with or without *Citrus reticulata* Blanco cv. Tankan peel essential oil (CPEO). Cell membrane fluidity (**C**) and integrity (**D**) of LM treated with or without CPEO. Columns marked with different letters imply that the differences are statistically significant (*p* < 0.05).

**Figure 4 foods-13-03841-f004:**
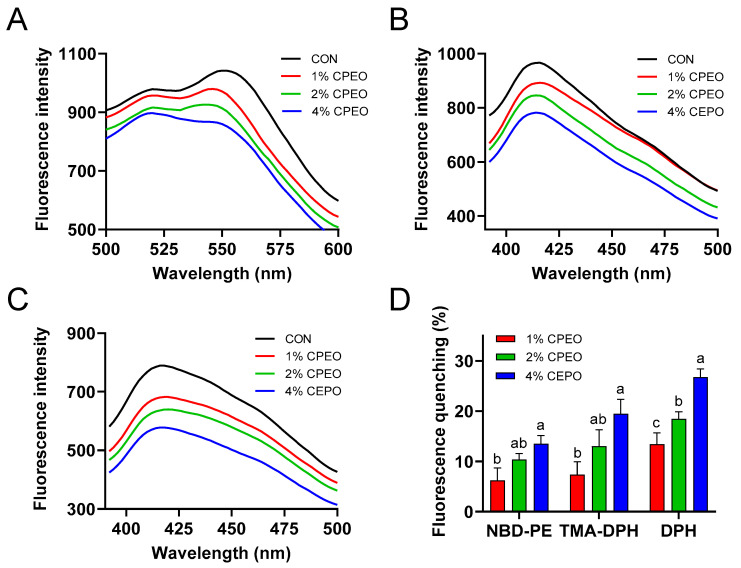
Exploring the effects of *Citrus reticulata* Blanco cv. Tankan peel essential oil (CPEO) on the cell membrane structure by using in vitro POPG liposome. Effects of CPEO on fluorescent spectrum of liposome tagged with various probes including NBD-PE (**A**), TMA-DPH (**B**), and DPH (**C**). The fluorescence quenching of various probes after CPEO treatment (**D**). Columns marked with different letters imply that the differences are statistically significant (*p* < 0.05).

**Figure 5 foods-13-03841-f005:**
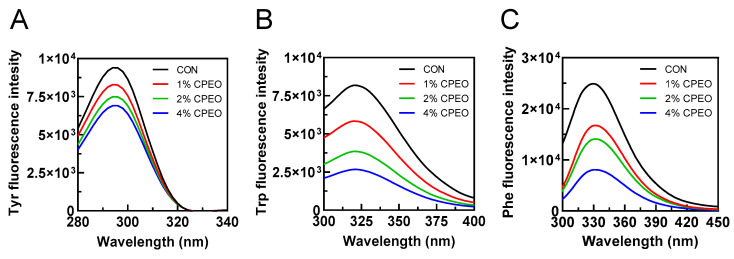
The membrane protein conformation of LM treated with or without *Citrus reticulata* Blanco cv. Tankan peel essential oil (CPEO). The variations in the fluorescent spectrum of the particular amino acids include Tyr (**A**), Trp (**B**), and Phe (**C**) in the membrane protein.

**Figure 6 foods-13-03841-f006:**
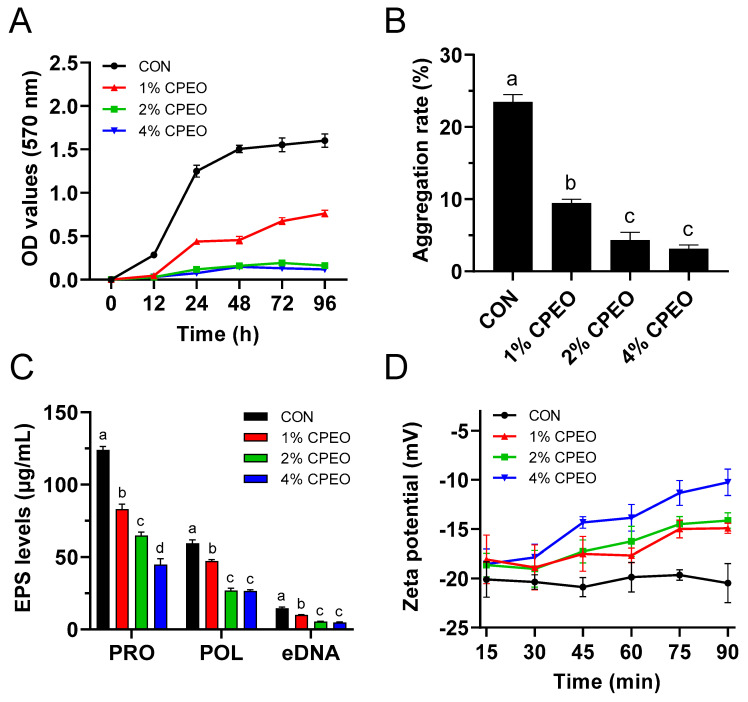
Biofilm formation curves of LM treated with or without *Citrus reticulata* Blanco cv. Tankan peel essential oil (CPEO) (**A**). Aggregation ability of LM treated with or without CPEO (**B**). Influences of CPEO on EPS (protein, PRO; polysaccharide, POL; eDNA) production of LM biofilm (**C**). Effects of CPEO on the surface charge of LM (**D**). Columns marked with different letters imply that differences are statistically significant (*p* < 0.05).

**Figure 7 foods-13-03841-f007:**
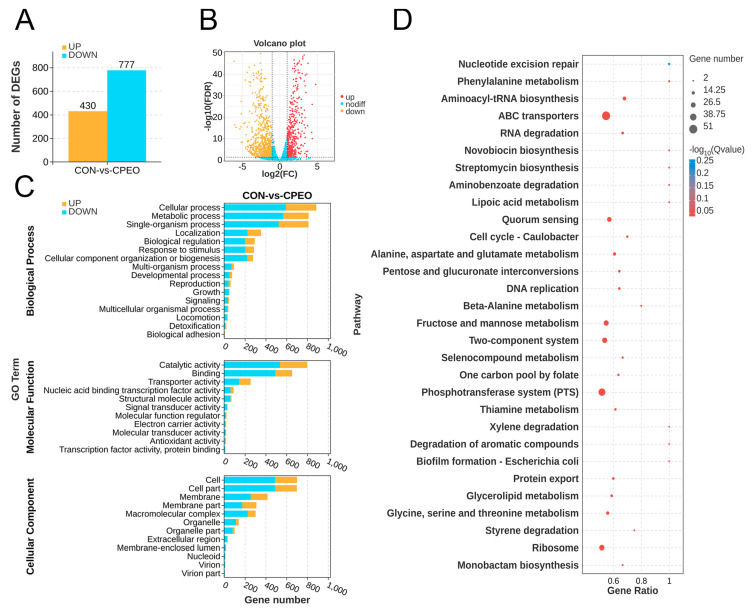
Transcriptomic profiles of *Citrus reticulata* Blanco cv. Tankan peel essential oil (CPEO)-treated LM in comparison to the control group (CON). (**A**) Numbers of up-regulated and down-regulated differentially expressed genes (DEGs). (**B**) Volcano plot analysis of DEGs in LM. (**C**) GO enrichment analysis categorizing DEGs into molecular function (MF), biological process (BP), and cellular component (CC). (**D**) KEGG enrichment bubble chart of DEGs. The larger the bubble, the greater the quantity, and the redder the bubble, the lower the P-value/Q-value ratio.

**Figure 8 foods-13-03841-f008:**
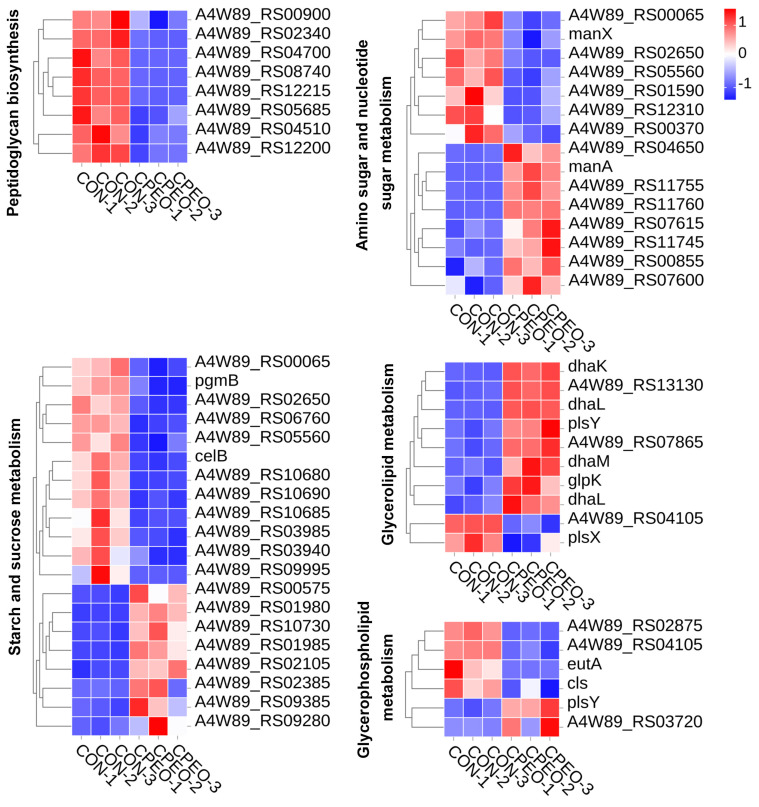
Heatmaps depicting representative DEGs related to cell wall synthesis and cell membrane integrity in LM, including peptidoglycan biosynthesis, amino sugar and nucleotide sugar metabolism, sucrose metabolism, glycerolipid metabolism and glycerophospholipid metabolism. Red denotes gene up-regulation while blue denotes gene down-regulation; the darker the color, the greater the change.

**Figure 9 foods-13-03841-f009:**
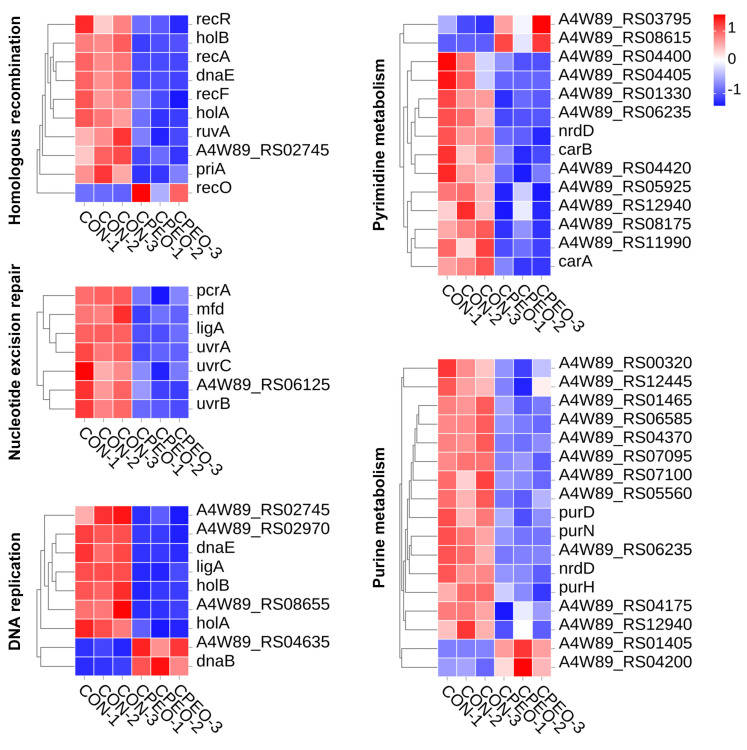
Heatmaps depicting the representative DEGs related to DNA replication and repair in LM. Red denotes gene up-regulation while blue denotes gene down-regulation; the darker the color, the greater the change.

**Figure 10 foods-13-03841-f010:**
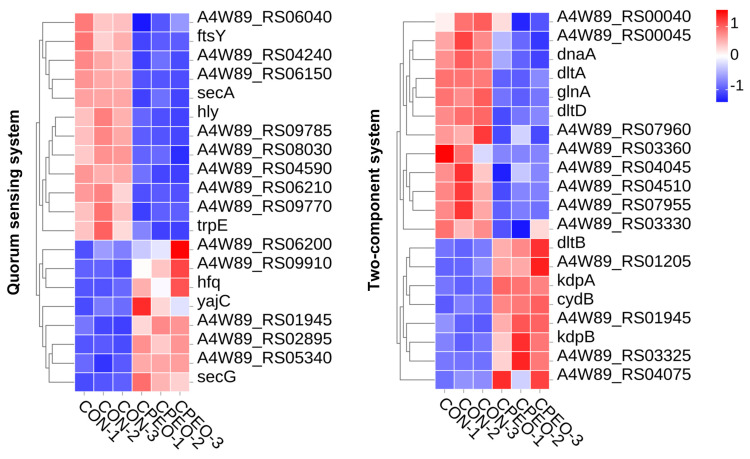
Heatmaps depicting the representative DEGs related to the quorum sensing system and two-component system in LM. Red denotes gene up-regulation while blue denotes gene down-regulation; the darker the color, the greater the change.

**Table 1 foods-13-03841-t001:** Chemical compositions of *Citrus reticulata* Blanco cv. Tankan peel essential oil (CPEO).

Peak No.	RT (min)	Compounds	CAS No.	Percentage (%)
1	7.930	α-pinene	007785−26−4	0.953% ± 0.032%
2	9.084	α-sabinene	003387−41−5	0.158% ± 0.004%
3	9.605	β-myrcene	000123−35−3	2.689% ± 0.121%
4	9.984	α-phellandrene	000099−83−2	0.056% ± 0.001%
5	10.779	limonene	005989−27−5	95.993% ± 3.763%
6	11.319	α-fenchene	000471−84−1	0.151% ± 0.005%

## Data Availability

The original contributions presented in this study are included in the article; further inquiries can be directed to the corresponding author.

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
