# Peer review of "Anti-Bacterial and Anti-Biofilm Activities of Essential Oil from *Citrus reticulata* Blanco cv. Tankan Peel Against *Listeria monocytogenes"

_foods, 2024, doi:10.3390/foods13233841_

Round 1
Reviewer 1 Report
Comments and Suggestions for Authors
Abstract:
Line 16-29: Citrus reticulata and Listeria monocitogenes are abbreviated as CM and LM respectively. This is not a common way of abbreviating Latin names and therefore I believe that such abbreviations should be avoided both in the abstract and throughout the text. The Latin names of the species should be abbreviated as usual: C. reticulata and L. monocitogenes.
Line 19-20: If the authors state in the abstract the methods they used in their research (which I think is appropriate), these must be stated fully and precisely, which is not the case here.
Line 28-29: In my opinion, the authors need to better explain the meaning and perspective of the results presented.
Keywords:
Line 30-31: In addition to the numerous methods the authors mention in the paper, the keywords are very sparse and based on what can be read from the title.
Introduction:
The English needs to be improved. The clarity and simplicity of expression contribute significantly to the quality of the text and the clarity of the results presented. Some sentences are unfinished (line 52-53, the problem probably lies in the linguistic expression ). Some sentences (line 55-58) are repetition from Abstract, without meaningfull explanation or better explanation of previous results.
In my opinion, the introduction lacks a better overview of the results to date as well as the linking of the existing literature results with the techniques used and the significance of the chosen approach to the problem.
Line 59-60: Sentence needs to be reworded.
Line 62: The full meaning of EPS should be given when it is mentioned for the first time in the main text. The abbreviation in the introduction is not sufficient (in my opinion it is not even necessary in the summary), especially when so many unusual abbreviations are used in the text.
Materials and Methods
A general remark that applies to most of the methods described: more experimental details should be included (everywhere and especially where the authors do not refer to a method described in detail in their previous work). The experiment must be clearly presented and therefore reproducible, which is impossible if not all important experimental details are given, e.g. whether the plant material used was dry or fresh, the amount of material is not clear (0.5 g seems to be a very small amount). Furthermore, it is not clear how the randomized sample was obtained and how many replicates were included. For the GC-MS analysis, it is not stated how the comparison of retention indices was made and whether the authors used a bank of mass spectra, how the percentage of components in the essential oil composition was determined and whether it was a triplicate analysis. Abbreviations that are not commonly used are listed again (e.g. BHI, EPS..), if the authors use them, they should explain them when they are used for the first time. It must also be indicated what the various time points mean (line 89); kit that are used must be specified (e.g. Cell Alkaline Phosphatase Activity Assay Kit, Solarbio, China), transcriptome analysis lack important details for further results interpretation.
Results and Discussion
It is customary to indicate the efficiency of the extraction, i.e. the amount of essential oil obtained and the total amount of constituents identified. The identified compounds can be sorted by structural groups and the retention indices are usually reported in the GC-MS results. The standard deviation of the triplicate analysis is also missing in Table 1. The libraries and standards used to identify the compounds should be listed in Material and Methods or below Table 1.
Results and discussion should follow Material and Methods, e.g. in Chapter 3.3. Bacterial morphology refers to the analysis of bacteria using TEM and SEM microscopes. The titles of Material and Methods and Results and Discussion should be linked to make the text more readable.
he results should generally be discussed in more detail in the context of existing or similar results. The discussion in connection with the results serves this purpose, so such a connection is necessary. Markers A and B are missing in Figure 10.
Comments on the Quality of English LanguageI give an example of the improvement of linguistic expression, which should be applied not only to the introduction but to the whole text. Clarity of expression contributes significantly to the readability of the text and thus to understanding the meaning of the results.
Listeria monocytogenes (LM) is one of the four major foodborne pathogens that is widely distributed in various contaminated foods, such as seafood, meat, raw milk, fruits and vegetables [1]. LM can form biofilms to enhance their viability in unfavorable environments, such as low temperatures, nutrient deficiencies and pH changes [2]. Consumption of food contaminated with LM can lead to listeriosis, a serious foodborne disease with a high mortality rate (around 30%) that particularly affects vulnerable populations such as the elderly, pregnant women and people with health problems [3]. Therefore, the development of anti-LM agents is of great importance to ensure food safety and public health. Since the use of synthetic antibacterial factors easily leads to bacterial resistance and environmental pollution [4], natural anti-LM substances such as polyphenols, essential oils, and peptides have recently gained much attention due to their multiple benefits [5-7].
Citrus fruits are among the most processed fruits worldwide and generate large amounts of waste such as peels, seeds and membrane residues every year [8]. Citrus peel waste accounts for almost 50-55% of the wet weight of the whole fruit [9]. Currently, citrus peel essential oils have attracted much attention due to their high yield, strong aroma and especially their broad-spectrum antibacterial properties [10-11]. Citrus essential oils are often used as a natural additive in food processing as they have been classified as safe by the FDA [12]. ……….? [13]. As with other citrus fruits, the main by-product of Citrus reticulata Blanco cv. Tankan is often discarded or used as animal feed, fertilizer and fuel. Previous studies have shown that the essential oils from the peel of Citrus reticulata Blanco cv. Tankan peel essential oils ( CPEO) had antibacterial abilities in Staphylococcus aureus and Escherichia coli [13]. However, the antibacterial and anti-biofilm activities of CPEO on LM have not yet been investigated.
Therefore, the antibacterial activities of CPEO against LM were investigated by evaluating its influences on cell morphology, membrane properties and membrane protein conformation. In addition, the effects of CPEO treatment on LM biofilm were investigated by measuring biofilm biomass, bacterial aggregation ability, EPS production and bacterial surface charge. Finally, transcriptome analysis was performed to clarify the anti-LM mechanism of CPEO at the genetic level.
Author Response
Dear Reviewer:
On behalf of myself and my co-authors, thank you for your efficient work in processing our manuscript entitled “Anti-bacterial and anti-biofilm activities of essential oils from Citrus reticulata Blanco cv. tankan peel against Listeria monocytogenes” (foods-3315768). The reviewers provided positive and constructive comments and suggestions that helped us improve our manuscript.
We made almost all the changes suggested by the reviewers and the editor. We checked the whole manuscript carefully and corrected spelling and language mistakes throughout the paper.
Revised portion are marked in red so that you can see all the changes. We provide detailed responses to all the reviewers’ comments below.
We hope that you agree that these changes make our paper acceptable for publication.
Sincerely,
Jun Li
Guangdong Key Laboratory of Science and Technology of Lingnan Specialty Food
Zhongkai University of Agriculture and Engineering
Guangzhou, China, 510225

Reviewer 2 Report
Comments and Suggestions for Authors
Well done, correct your manuscript according my comments in attached revised manuscript

Author Response

(The authors gave the same response as above.)

Reviewer 3 Report
Comments and Suggestions for Authors
The authors have prepared a detailed and explanatory article regarding the anti-bacterial properties of essential oils from Citrus reticulata species peel against Listeria monocytogenes.
Please take into account the following:
1. start your abstract in a more introductive way
2. line 50-51: "Citrus essential oils are..." and similar grammar check
3. in my opinion would be more useful to divide the results from disscusion. Moreover, even if you insist to keep only conclusions seperated, your disscusion needs to be more extrnsive and try to include more articles that study the excact EO, its antimicrobial properties (even with other bacteria) and its transcriptome profile. More cocnclusions will be conducted this way. Additionally, a comparison of the major constituent of these similar EO will be beneficial.
Author Response

(The authors gave the same response as above.)

Round 2
Reviewer 1 Report
Comments and Suggestions for Authors
Line 17-20:
In my opinion, the sentence should be rephrased in accordance with what the authors wanted to emphasise (Citrus reticulata Blanco cv. Tankan peel essential oil (CPEO) showed anti-Listeria monocytogenes (LM) activities, and the authors investigated the associated mechanisms using different techniques):
The aim of this study was to investigate the anti-Listeria monocytogenes (LM) activities of essential oils from the peel of Citrus reticulata Blanco cv. Tankan (CPEO) and the associated mechanism using a high-resolution electron microscope, fluorescence spectrometer, flow cytometer, potentiometer, and transcriptome sequencing.
Line 17
Citrus reticulata Blanco cv. tankan should be corrected as Citrus reticulata Blanco cv. Tankan (also through the text)
Line 28-30:
These results indicated that CPEO could be served as a novel anti-LM agent and green preservative in the food field. Further study is needed to verify the anti-LM activities of CPEO in real food.
I suggest improvement of the sentence as follows:
These results suggest that CPEO could be used as a novel anti-LM agent and green preservative in the food sector. Further studies are needed to verify the anti-LM activities of CPEO in real food.
Line 222
The standard deviation of the triplicate analysis (?) is still missing in Table 1.
Author Response

(The authors gave the same response as above.)
